# *Aedes albopictus* microbiota: Differences between wild and mass-reared immatures do not suggest negative impacts from a diet based on black soldier fly larvae and fish food

Carlo Polidori[1]*, Andrea Ferrari[1], Luigimaria Borruso[2]*, Paola Mattarelli[3], Maria Luisa Dindo[3], Monica Modesto[3], Marco Carrieri[4], Arianna Puggioli[4], Federico Ronchetti[5,6], Romeo Bellini[4]

1 Department of Environmental Science and Policy (ESP), University of Milan, Via Celoria, Milan, Italy,
2 Faculty of Agricultural, Environmental and Food Sciences, Free University of Bolzano, Piazza Università, Bolzano, Italy, 3 Department of Agricultural and Food Sciences, University of Bologna, Viale G. Fanin, Bologna, Italy, 4 Centro Agricoltura Ambiente "G. Nicoli", IAEA Collaborating Centre, Via Sant'Agata, Crevalcore, Italy, 5 Department of Biosciences and Pediatric Clinical Research Center "Romeo and Enrica Invernizzi", University of Milan, Milan, Italy, 6 Italian Malaria Network, Inter University Center for Malaria Research, University of Milan, Milan, Italy

* carlo.polidori@unimi.it (CP); luigimaria.borruso@unibz.it (LB)

## Abstract

The "Sterile Insect Technique" (SIT), a promising method to control *Aedes albopictus*, the Asian tiger mosquito, is gaining increasing interest. Recently, the role of microbiota in mosquito fitness received attention, but the link between microbiota and larval diet in mass rearing programs for SIT remains largely unexplored. We characterized the microbiota of four larval instars, pupae and eggs of non-wild (NW) lab-reared *Ae. albopictus* fed with a diet based on Black soldier fly (*Hermetia illucens*) larvae powder and fish food KOI pellets. We compared it with wild (W) field-collected individuals and the bacterial community occurring in rearing water-diet (DIET). A total of 18 bacterial classes with > 0.10% abundance were found overall in the samples, with seven classes being especially abundant. Overall, the microbiota profile significantly differed among NW, W and DIET. Verrucomicrobiae were significantly more abundant in W and DIET, Bacteroidia were more abundant in NW and DIET, and Gammaproteobacteria were only more abundant in W than in DIET. W-eggs microbiota differed from all the other groups. Large differences also appeared at the bacterial genus-level, with the abundance of 14 genera differing among groups. Three ASVs of *Acinetobacter*, known to have positive effects on tiger mosquitoes, were more abundant in NW than in W, while *Serratia*, known to have negative or neutral effects on another *Aedes* species, was less abundant in NW than in W. The bacterial community of W-eggs was the richest in species, while dominance and diversity did not differ among groups. Our data show that the diet based on Black soldier fly powder and fish food KOI influences the microbiota of NW tiger mosquito immature stages, but not in a way that may suggest a negative impact on their quality in SIT programs.

**Data Availability Statement:** All relevant data are within the paper and its Supporting Information files.

**Funding:** The author(s) received no specific funding for this work.

**Competing interests:** The authors have declared that no competing interests exist.

## Introduction

Adult females of most mosquito species (Diptera: Culicidae) need a blood meal from a vertebrate host to produce and lay eggs [1]. The blood-feeding behaviour is exploited by several pathogens causing human and animal diseases [2]. *Aedes*, *Culex* and *Anopheles* are the genera most associated with spreading pathogenic agents [3]. Mosquito-borne diseases are causing high morbidity and mortality worldwide (Global Health Estimates (who.int)). Socio-economic factors such as urbanization and globalization of trade, along with changes in climate conditions, are facilitating the spread of mosquito vectors beyond their native ranges [4]. This is the case of *Aedes albopictus* Skuse, 1894, the Asian tiger mosquito, native to Southeast Asia and now common in Europe [5]. This species is a highly efficient vector of several arboviruses to vertebrates, including humans [6].

Due to the increasing public health concern coupled with the relevant nuisance related to the anthropophilic behaviour of *Ae. albopictus*, a plethora of population control approaches have been developed; including environmental, mechanical, biological, chemical, and genetic methods [7]. However, currently available control approaches were insufficient in reducing local population levels of the tiger mosquito [7–10]. Among the newly proposed mosquito control methods, the "Sterile Insect Technique" (SIT) is emerging as one of the most promising. It consists in mass-rearing a target species followed by the selection and sterilization of the male individuals using ionizing radiation [11]. After sterilization, males are released in the environment where they will likely mate with virgin wild females of the same species making them unable to lay fertile eggs. This method brings several advantages because it is species-specific, environmentally non-polluting and has been proven effective in reducing the target wild population (*Ae. aegypti* and *A. albopictus*) after regular releases of sterile males [11].

However, producing sterile males of high quality (i.e., competitive with wild ones) while limiting production costs, remains challenging [12,13]. For this purpose, the role of microbiota in developing more efficient mass-rearing aimed at SIT application has recently received increasing attention [14,15].

In mosquitoes, the gut microbiota is known to heavily affect the health and fitness of the host [16–18], and the largest part of this microbiota is acquired through the rearing-water diet in which the larvae develop [14,19]. Controlled experiments showed how *Anopheles* and *Aedes* larvae might transmit a portion of their gut microbiota to adults through metamorphosis [20]. Mosquitoes' gut microbiota consists primarily of bacteria and, to a lesser extent, of fungi and algae. The largest part of the bacteria community comprises four phyla (Pseudomonadota, Bacillota, Bacteroidota and Actinomycetota) [16] and thus accounting for an overall low diversity. In fact, mosquito microbiota comprises around 200 species of mostly gram-negative aerobic species [21].

Mosquitoes probably rely on variable bacteria communities acquired through the aquatic larval habitat rather than on specific bacterial taxa [22]. Indeed, it has been shown that bacterial diversity and gut microbiota varies between and within species and are affected by several factors [19]. In particular, the bacterial community occurring in aquatic habitats plays a significant role in shaping the gut microbiota [23]. Hence, the environment, rather than genetics, shapes the gut microbiota of mosquitoes, as in many other insects that acquire their gut microbiota from the environment [24,25]. Such water-derived microbiota makes crucial the implementation of adequate larval diets in mass-rearing programs. In light of the central role of the environment in shaping microbiota, the study of the tripartite interaction network among larvae, diet and the microbiota could represent a key tool to obtain the best mass-rearing conditions through an optimization of the larval medium.

A promising research line could be the development of novel diets, such as those containing probiotics supplements, to improve larval survival, adult longevity, dispersal ability or sexual

performance [15]. However, the role of the diet and the development of new media to improve current SIT methodologies is still largely overlooked [26–28], especially regarding the effects of such diets on mosquito microbiota. Thus, the aim of this work was to characterize the bacterial microbiota of the eggs, four larval instars and pupae of lab-reared *Ae. albopictus* fed with a novel diet based on Black soldier fly (*Hermetia illucens* [Linnaeus, 1758]) (Diptera: Stratiomyidae) larvae powder and fish food KOI powder. To evaluate the effects of this diet, we compared the microbiota of the lab-reared insects with the one of wild individuals and with the bacterial community occurring in the rearing water-diet. This will provide new insights on the possible effects of innovative diets on mosquito larvae microbiota, thus likely providing indication to improve the quality and performances of sterile males.

## Methods

### Laboratory and field sampling

<u>Lab reared mosquitoes and their diet</u>: a strain of *Ae. albopictus* reared under controlled laboratory conditions for several generations (strain Bologna F8) has been studied. This lab-reared sample is referred as NW (non-wild) in the following text. Mosquito eggs were obtained following standard rearing procedures developed at the "Centro Agricoltura e Ambiente" (CAA) [29,30]. Filter papers with one-week-old eggs were gently brushed off, and eggs were stored inside closed plastic boxes with a saturated potassium sulfate solution to maintain a high humidity level. Upon hatching, first instars (I instar larva, L1) were reared with a density of 2 larvae/ml (i.e., 4000 larvae in 2 l water) in a climate chamber at 28°C, 80% RH and 14:10 L:D. Larvae and pupae were collected with a pipette, rinsed, and transferred into sterile 50 ml Falcon tubes in water. Microbiota analysis was conducted on samples of eggs, I-II-III-IV instar larvae (L1, L2, L3 and L4), and pupae (S1 Table).

For what concerns the diet, the formulation used for this study consisted of Black soldier fly larvae powder (InnovaFeed, Évry, France; http://www.innovafeed.com/) and fish food KOI pellets (KOI-Franciacorta, BS, Italy) finely grounded (50:50) and mixed with water [26,28,31,32]. Larvae were daily supplied with food, in the form of a slurry (3% w:v), until pupation occurred. Such a diet was previously shown to be suitable for the development and quality of the produced insects in terms of time to pupation, adult production, and male flight ability [28]. Rearing water-diet samples of 5 ml, each were collected with a pipette from the larval rearing tray corresponding to each development stage and transferred into sterile 15 ml Falcon tubes. Microbiota analysis was conducted on samples of five rearing water, such as DIET_L1, DIET_L2, DIET_L3, DIET_L4 and DIET_pupae, where L1, L2, L3, L4 and pupae were grown.

<u>Wild mosquitoes:</u> to determine the diversity in microbiota of wild *Ae. albopictus* in respect to lab reared mosquito, samples were collected from catch basins at Bologna (Emilia Romagna region, Italy) in October 2020. Samples of L1, L2, L3, and L4 and pupae were collected at two sites (Site A: Lat 44°29'14.94"N–Long 11°16'52.34"E; Site B: Lat 44°32'2.90"N–Long 11°20'41.01"E). Eggs were collected using ovitraps alongside the larval habitats. We refer to the sample collected in the field as W (wild) in the following text.

For the detailed description of samples and sampling sites see S1 Table.

All samples, weighing at least 25 mg for most of them (except for field-collected samples where the availability was very low) (S1 Table), were preserved in a -80°C freezer. S1 Table describes the correspondence between weight and the number of larvae, pupae, and eggs tested.

### DNA extraction and amplicon sequencing

For all samples (W and NW, together with rearing water, S1 Table) three replicates were used for DNA extraction.

Before DNA extraction, the collecting Falcon tubes were centrifuged at 6°C for 30 min at 8,000 rpm. The supernatant was discarded, and 20 mL EtOH was added before another centrifugation at 6°C for 10 min. The supernatant was discarded, and the tubes were kept at 65°C for 10 min to evaporate the remaining EtOH. 50mg of each sample, when possible (S1 Table), were utilized for DNA extraction according to the Dneasy Blood and Tissue protocol (QIAGEN, Hilden, Germany). DNA extraction was performed for the DIET samples using the Dneasy PowerSoil Kit (QIAGEN, Hilden, Germany). Isolated DNA concentration and purity (absorbance ratio 260/280 and 260/230) were checked by spectrophotometry using NanoDrop (Fisher Scientific, 13 Schwerte, Germany).

The V3-V4 region of the 16S rRNA gene (~ 460 bp) was amplified and sequenced using the Illumina MiSeq platform 300x2bp. Gene amplicons were produced using the primers Pro341F: `5′-TCGTCGGCAGCGTCAGATGTGTATAAGAGACAGCCTACGGGNBGCASCAG-3′` and Pro805R: `5′GTCTCGTGGGCTCGGAGATGTGTATAAGAGACAGGACTACNVGGGTAT CTAATCC-3′` [33], using Platinum™ Taq DNA Polymerase High Fidelity (Thermo Fisher Scientific, Italy). The libraries were prepared following [34].

All sequences have been submitted to the European Nucleotide Archive (EMBL-EBI) under the accession number for SRA data PRJNA949646 (Temporary Submission ID: SUB12996474). SRA records will be accessible with the following link after the release date (2024-05-30): https://www.ncbi.nlm.nih.gov/sra/PRJNA949646.

### Statistical analysis

The raw sequences data were quality-checked through FastQC [35]. Next, sequences data were preprocessed, quality filtered, trimmed, de-noised, merged and modelled via DADA2 [36] within QIIME2 [37]. Chimeras were discarded according to the 'consensus' method [36]. Representative sequences variants (ASV) were taxonomically assigned using a Naïve–Bayes classifier trained Silva 138.

All statistical analyses were based on the cleaned ASV matrix provided in the Supporting file DATASET.xls. First, we explored the microbiota composition of DIET, eggs, L1, L2, L3, L4 and pupae by calculating the average abundances of the bacterial classes. Statistical tests were not applied to evaluate differences among mosquito stages due to the small sample size within groups (see above). Instead, we applied statistics to evaluate differences among three groups: DIET (n = 5), W (n = 9) and NW (n = 6), regardless of stage.

First, we analyzed the ASVs table using a Bray-Curtis dissimilarity matrix as a suitable distance measure for zero-inflated data [38]. These analyses do not require *a priori* grouping of species, meaning that these methods allow pattern formation that is exclusively based on microbiota similarities. We first performed an agglomerative cluster analysis based on the unweighted pair group method using arithmetic means of Bray–Curtis dissimilarities. Second, Bray-Curtis dissimilarities were used for ordinations using non-metric multidimensional scaling analysis (NMDS), which is a non-parametric method that avoids assuming linearity among variables [39] and whose resulting plot shows the spatial distances between individuals (i.e. their microbiota distances). In the NMDS, deviations are expressed in terms of "stress", for which values $\leq 0.15$ indicate a good fit of ordination [40]. ANOSIM (Non-Parametric Analysis of Similarity) was employed to test for differences among the three groups. The significance is computed by permutation of group membership (9999 replicates). Pairwise ANOSIM between all pairs of groups was also computed as a post-hoc test. Similarity percentages (SIMPER) were calculated to identify the bacterial taxa that predominantly contributed to the Bray-Curtis dissimilarities among pairs of species [41].

Second, we tested more in detail differences in the abundance of the most abundant bacteria classes and genera (> 1% abundance in both cases) among the three groups with Kruskal-

Wallis tests, followed by paired comparisons through Dunn's procedure. Kruskal-Wallis tests (and Dunn's paired tests) were also used to compare four measures of diversity: richness (*S*) (total number of ASVs), diversity (*H*) (Shannon-Weaver Index, ranging from 0 for microbiota with only a single taxon to high values if many taxa, each with few individuals, occur), dominance (*D*) (i.e. 1-Simpson index, ranging from 0 (all taxa are equally present) to 1 (one taxon dominates the community completely)) and evenness (E) (i.e. ASVs distribution (*H*/log(*S*)), ranging from 0 to 1), among the three groups.

All analyses were carried out in PAST 4.03 (Paleontological Statistics Software Package) [42]. In all results and tables, mean values are expressed ± standard error.

## Results

The data about DNA yield from each sample have been described in S1 Table: the amount of DNA obtained ranged from 47.54±6.44 (ng/ul). Raw sequence data consisted of 1,141,408 reads with an average of 57,070 ±19,223 per sample. After bioinformatics pipelines and quality filtering, a total of 776,170 bacterial reads were found, with an average of 38,808 ± 13,544 per sample. The reads grouped into 2,400 with an average of 201 ± 153 per sample. Rarefaction curves showed that all the samples approached saturation (S1 Fig). A total of 18 bacteria classes with overall > 0.10% abundance across the samples were found in the microbiota (Fig 1).

Gammaproteobacteria, Bacteroidia, Alphaproteobacteria, Actinobacteria, Bacilli, Spirochaetia and Verrucomicrobiae were the most abundant classes (> 1%) (Fig 1). However, their relative importance clearly differs among W, NW and DIET. While Verrucomicrobiae were significantly more abundant in W and DIET, Bacteroidia were more abundant in NW and DIET (Table 1). On the other hand, Gammaproteobacteria did not differ between W and NW but were more abundant in W than in DIET (Table 1). Though not significantly, also Actinobacteria were more abundant in NW and DIET (Table 1). An entire class, Spirochaetia, was only found in W sample (Table 1), and only in L4. Differences appear also considering bacterial families (S2 Fig). For example, NW samples include great abundances of Weeksellaceae and Moraxellaceae, while W samples show Yersiniaceae as an abundant family (S2 Fig).

The cluster analysis based on the abundance of all ASVs reflects such large scale (e.g. classes) differences (Fig 1), with three main clusters including respectively all samples of W, NW and DIET (the latter closer to NW and more distant to W). The only exception was the W-eggs, which formed a cluster of their own, distant from the other W-samples (Fig 1).

Considering more in detail the bacterial genera with > 1% abundance (for a total of 21 genera), further differences emerged among the three groups (S2 Table, Fig 2).

Overall, 14 genera (some including more than one ASVs) differed among groups. In particular, four genera (one occurring in two strains) were significantly more abundant in DIET compared with W and NW: *Acinetobacter*, *Flavobacterium*, *Comamonas* and *Sphingobacterium* (S2 Table, Fig 2 black box). On the other hand, *Elizabethkingia* and three further ASVs of *Acinetobacter* were much more abundant in NW than in W and DIET (S2 Table, Fig 2 blue box). Finally, four genera (three determined only at family level, with one occurring in two ASVs) essentially only occurred in W: *Serratia*, *Comomonadaceae* (two ASVs) and *Methylophilaceae* (uncultured) (S2 Table, Fig 2 red box). Interestingly, the endosymbiont *Wolbachia*, albeit occurring at > 1% abundance, did not differ between W and NW and was absent in DIET (S2 Table). All the other genera among the 21 more abundant ones did not show significant differences among the three groups (S2 Table).

The NMDS shows that the microbiota profile significantly differs among W, NW and DIET (stress = 0.15, ANOSIM: Mean rank within = 50.1, Mean rank between = 116.9, $R^2$ = 0.7,

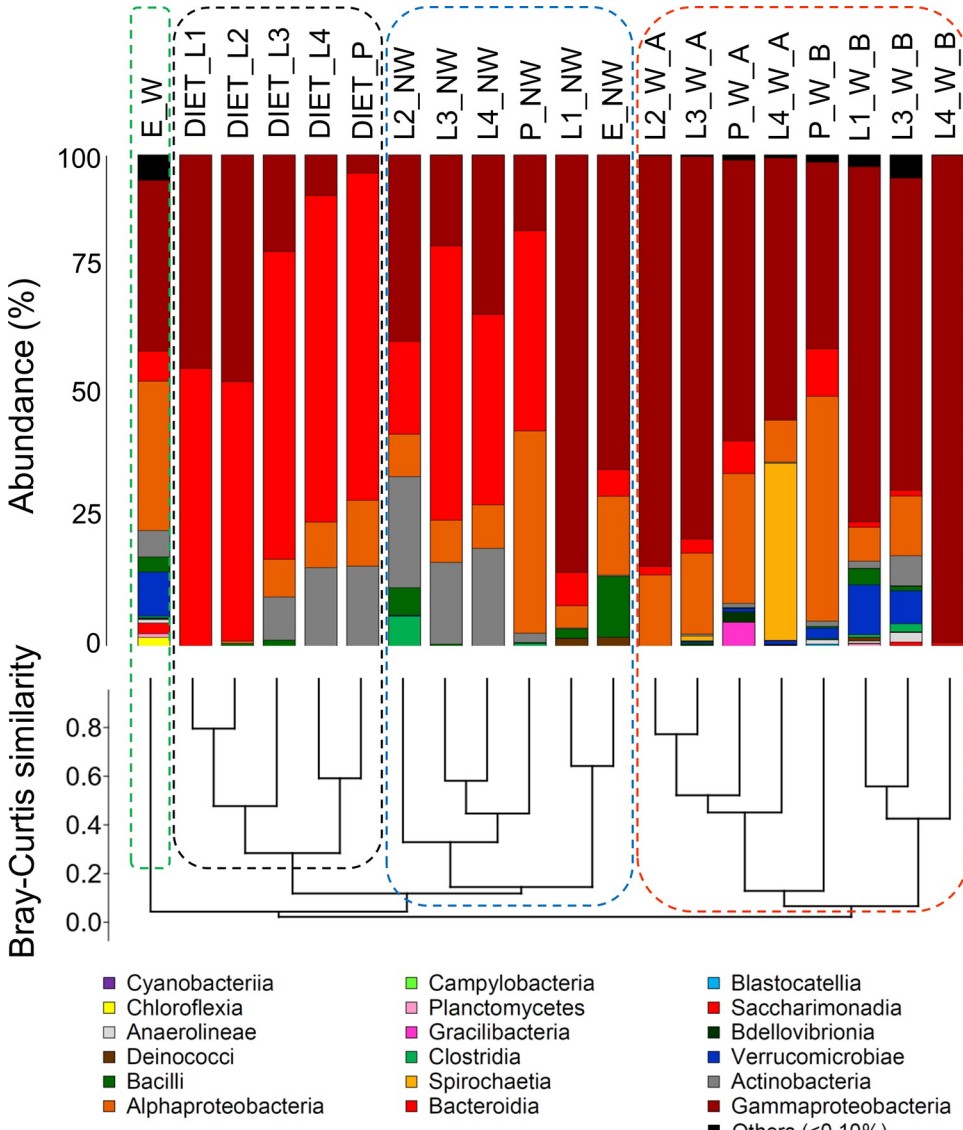

**Fig 1. Above, stacked barplots showing the relative abundance (%) of the bacterial classes; below, dendrogram representing the hierarchical clustering based on Bray-Curtis similarity.** Dashed lines connect the two graphs and group the four clusters emerged from the Bray-Curtis similarity index (dashed lines: Green = E; blue = diet; black = NW; red = W). Bacterial classes accounting for less than 0.1% of the total abundance were grouped in the category "other". L: Larvae, P: Pupae, E: Eggs. W: Sample collected in the field (wild); NW: Lab-reared sample (non-wild).

P < 0.0001) (Fig 3), with Bray-Curtis distance greatest between W and DIET (99.1) and lowest between DIET and NW (88.2).

All paired differences were significant (0.0002 < P < 0.0022). Despite the small sample, a difference between W localities is also apparent, while less clear is the effect of mosquito developmental stage on microbiota profile (Fig 3). However, interestingly, eggs seem to possess a unique profile in W, but a profile similar to L1 in NW (Fig 3).

The SIMPER analysis shows that differences among the three microbiota profiles (W, NW and DIET) essentially depend on 21 abundant genera of bacteria which account for > 1% of dissimilarity (S3 Fig). In particular, *Elizabethkingia* (more abundant in NW), *Serratia* (only

**Table 1. Descriptive statistics (% mean values ± SE) of the 7 classes represented by more than 1% in the bacterial communities analysed, with statistical differences among groups (W, NW, DIET).**

| Class | W ($N = 9$) | NW ($N = 6$) | DIET ($N = 5$) | Kruskall-Wallis test | Dunn's paired comparisons |
|---|---|---|---|---|---|
| Actinobacteria | 1.94 ± 0.86 | 10.47 ± 4.45 | 8.40 ± 3.72 | $\chi^2 = 1.51$, $P = 0.45$ | - |
| Alphaproteobacteria | 19.08 ± 4.44 | 14.82 ± 5.54 | 6.16 ± 2.56 | $\chi^2 = 3.16$, $P = 0.20$ | - |
| Bacilli | 0.97 ± 0.49 | 3.68 ± 1.99 | 0.70 ± 0.28 | $\chi^2 = 2.7$, $P = 0.25$ | - |
| Bacteroidia | 3.56 ± 1.18 | 27.96 ± 8.25 | 60.71 ± 2.70 | $\chi^2 = 14.57$, $P = 0.0007$ | W vs. DIET: $P < 0.001$<br>W vs. NW: $P = 0.03$<br>NW vs. DIET: $P = 0.11$ |
| Gammaproteobacteria | 66.65 ± 6.65 | 43.03 ± 11.33 | 24.02 ± 8.69 | $\chi^2 = 6.47$, $P = 0.04$ | W vs. DIET: $P = 0.01$<br>W vs. NW: $P = 0.13$<br>NW vs. DIET: $P = 0.33$ |
| Spirochaetia | 4.18 ± 4.03 | 0.00 | 0.00 | $\chi^2 = 8.25$, $P = 0.005$ | W vs. DIET: $P = 0.018$<br>W vs. NW: $P = 0.003$<br>NW vs. DIET: $P = 0.68$ |
| Verrucomicrobiae | 3.62 ± 1.48 | 0.04 ± 0.03 | 0.01 ± 0.01 | $\chi^2 = 14.31$, $P = 0.0006$ | W vs. DIET: $P < 0.001$<br>W vs. NW: $P < 0.001$<br>NW vs. DIET: $P = 0.67$ |

Dunn's procedure for paired comparisons was performed only for significant overall difference among groups. W: Sample collected in the field (wild); NW: Lab-reared sample (non-wild).

present in W) and *Acinetobacter* (more abundant in NW) seem very important in discriminating NW from W. *Elizabethkingia*, *Sphingobacterium* (more abundant in DIET) and *Acinetobacter* seem important in discriminating NW from DIET, while *Serratia*, *Sphingobacterium* and *Acinetobacter* seem important in discriminating W from DIET (S3 Fig).

The bacterial community of different developmental stages was highly heterogeneous regarding species richness (*S*) and the three diversity indices (Table 2). In particular, the bacterial community of W-egg samples (E_W) resulted in being the richest in species (*S*), the lowest in dominance (*D*) and the highest in both Shannon index (*H*) and evenness (*E*) (Table 2). However, we did not find any statistically significant differences among the three groups in terms of *H*, *D* and *E* (Table 2). Only species richness differed among the three groups (Kruskal-Wallis test, $\chi2 = 8.566$, $P = 0.014$, Table 2). The post-hoc Dunnett's test showed that the DIET group had a poorer bacterial community than the W group ($P = 0.022$, Table 2) but no difference was found with the NW group ($P = 0.011$, Table 2).

## Discussion

We described the microbiota of *Ae. albopictus* reared in a diet based on Black-soldier fly larvae powder and fish food KOI powder and compared it with that observed in natural populations and in the diet medium itself. We found important differences in bacterial diversity between wild and non-wild individuals and their diets. Such differences point to a certain, though apparently weak, acquisition of bacterial components from the rearing diet to immature stages as shown by the lab-reared mosquitoes' microbiota profile. The occurrence of only a few bacterial taxa shared between mosquitoes and the diet suggests that the mosquitoes internal body environment allows the survival of only a few bacteria, as also noted by previous studies on *Aedes* [15,21,43,44].

Despite the relatively low diversity, the gut microbiota has been shown to play a central role both during larval and adult stages of different mosquito species. Obviously, we actually do not know if all the described effects are equally relevant and if similar functions are also showed by the bacteria found in our studied populations, but these previous investigations can certainly

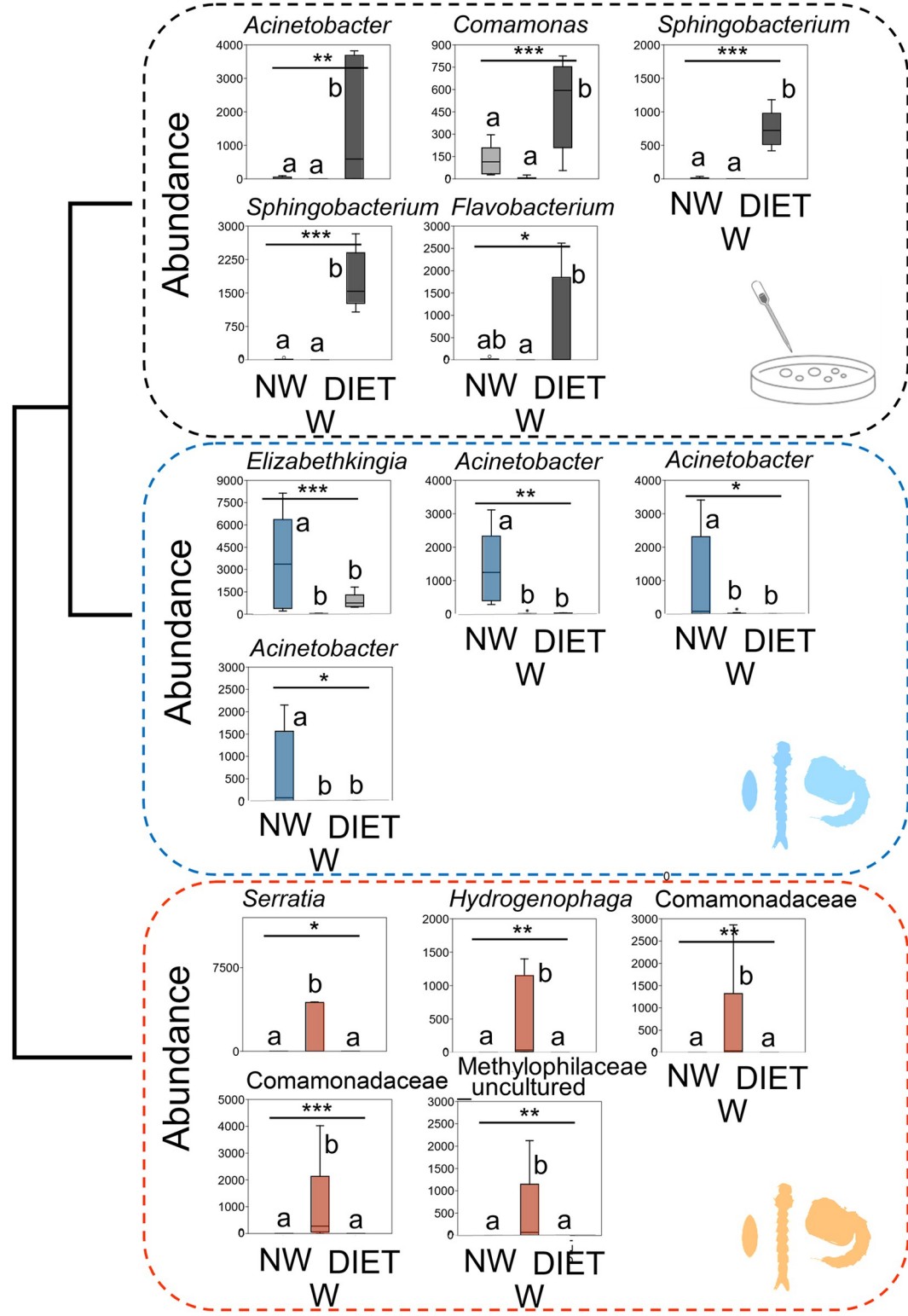

**Fig 2. Box-and-whisker plots showing medians (horizontal lines within boxes), 1° and 3° quartiles (horizontal lines closing the boxes), and maximum and minimum values (ends of the whiskers) of the abundances (%) of the bacterial genera accounting for > 1% of the total abundance and showing significant differences among groups Non-wildtype (NW), Wildtype (W) and DIET.** *Wolbachia* was excluded since, although covering > 1% abundance, it was absent from DIET and did not differ between W and NW. In the grey group, bacterial genera with higher abundances in DIET, in the

blue group, bacterial genera with higher abundances in NW, and in the red group, bacterial genera with higher abundances in W. The three groups are clustered (left side) as depicted by the hierarchical clustering (see Fig 1). Note that few bacterial genera appear in more than one Box-and-whisker plot, because they represent different SVs. Undetermined genera were named as the family they belong and referred as "uncultured" or "unknown". Different letters were used to show the results of the pairwise comparisons using Dunn's procedure. * P < 0.05, ** P < 0.01, *** P < 0.001.

encourage new studies on our model system to test for possible parallelisms. For example, Mitraka et al. [45] showed how bacteria of the genus *Asaia*, one of the principal members of *Anopheles* microbiota, promoted larval growth. *Asaia*, together with *Escherichia coli*, has also proven beneficial for *Ae. aegypti* (L., 1762) by increasing its longevity [46]. In the genus *Culex*, instead, bacteria of the genera *Klebsiella* and *Aeromonas* were shown to be a crucial food source for the most fragile instar (L1), assuring its development into a more resilient subsequent stage (L2) [47]. Bacteria also play a role in egg production and oviposition. Particularly, bacteria of the genus *Comamonas* supported development and egg production in an *Aedes* species [18], while bacteria of the genera *Klebsiella* and *Aeromonas* enhanced oviposition [47]. Of

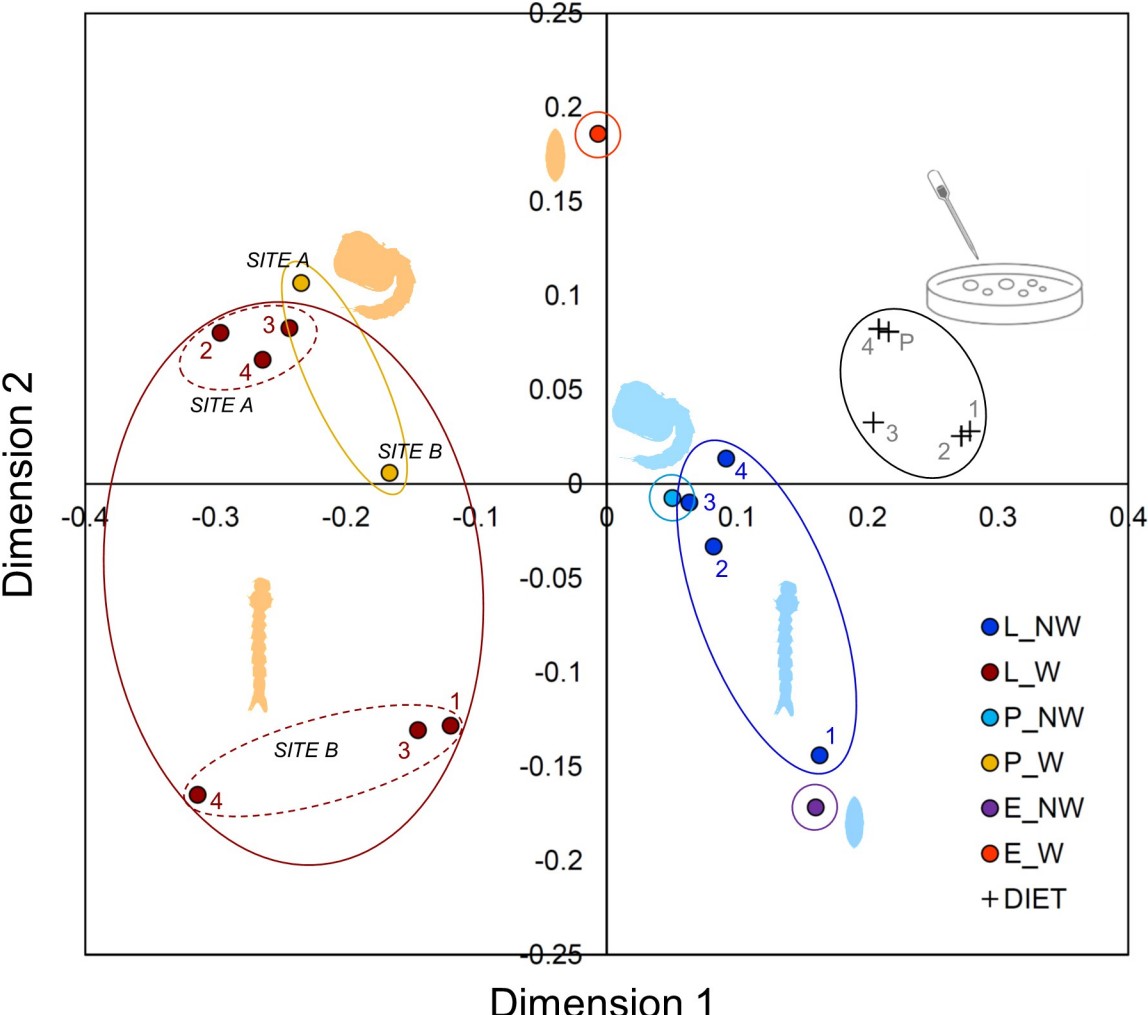

**Fig 3. Plot resulting from the Non-Metric Multi-Dimensional Scaling (NMDS) analysis of the bacterial composition in different mosquito immature stages and in diet.** L: Larvae, P: Pupae, E: Eggs. W: Sample collected in the field (wild); NW: Lab-reared sample (non-wild).

**Table 2. Richness (Taxa_S) and diversity indices for all the samples analysed in this study.**

| type | sample | Taxa_S | Simpson_1-D | Shannon_H | Evenness |
|---|---|---|---|---|---|
| **W** | L4_W_B | 62 | 0.44 | 1.11 | 0.19 |
| **DIET** | DIET_L2 | 83 | 0.90 | 3.01 | 0.47 |
| **DIET** | DIET_L1 | 85 | 0.90 | 2.91 | 0.45 |
| **NW** | L4_NW | 103 | 0.83 | 2.65 | 0.40 |
| **NW** | E_NW | 118 | 0.95 | 3.57 | 0.52 |
| **NW** | L1_NW | 124 | 0.90 | 3.09 | 0.44 |
| **NW** | P_NW | 127 | 0.80 | 2.49 | 0.36 |
| **NW** | L3_NW | 129 | 0.70 | 2.19 | 0.31 |
| **DIET** | DIET_P | 140 | 0.93 | 3.37 | 0.47 |
| **DIET** | DIET_L3 | 141 | 0.95 | 3.54 | 0.50 |
| **NW** | L2_NW | 151 | 0.92 | 3.35 | 0.46 |
| **DIET** | DIET_L4 | 158 | 0.94 | 3.48 | 0.48 |
| **W** | L2_W_A | 176 | 0.90 | 3.10 | 0.42 |
| **W** | L4_W_A | 202 | 0.85 | 2.84 | 0.37 |
| **W** | L3_W_A | 216 | 0.93 | 3.55 | 0.46 |
| **W** | P_W_A | 222 | 0.96 | 3.85 | 0.49 |
| **W** | P_W_B | 251 | 0.89 | 3.61 | 0.45 |
| **W** | L3_W_B | 331 | 0.87 | 3.28 | 0.39 |
| **W** | L1_W_B | 359 | 0.89 | 3.52 | 0.41 |
| **W** | E_W | 746 | 0.99 | 5.78 | 0.61 |

L: Larvae, P: Pupae, E: Eggs. W: Sample collected in the field (wild); NW: Lab-reared sample (non-wild). Data from field sampling sites A and B were pooled together.

increasing interest is also the role that the adult mosquito midgut microbiota could play in modulating pathogen transmission [48]. Studies showed how, in the genera *Anopheles* and *Aedes*, bacteria of the genera *Enterobacter* [49], *Serratia* [50] or *Wolbachia* [51] induced refractoriness to *Plasmodium* infection. Bacteria of the genus *Wolbachia* also play a role in controlling mosquito mating through cytoplasmic incompatibility [52]. Cytoplasmic incompatibility prevents infected males from producing viable progeny when mating with an uninfected female and could thus be exploited as a novel SIT technique [53].

Particularly interesting for our case study, three strains of *Acinetobacter*, known to have positive effects on mosquitoes, were more abundant in NW than in W. In particular, *Acinetobacter* seems to promote larval growth in *Ae. aegypti* [54], and it ensures a complete development in the larvae of *Stomoxys calcitrans* L., 1758 (Diptera: Muscidae) [55]. In *Ae. albopictus* females, *Acinetobacter* seems to improve blood digestion and nectar assimilation and possibly improve the capability to adapt to anthropized habitats [56]. *Elizabethkingia*, which we found more abundant in NW than W, is known to have a positive effect on some *Anopheles* species. In fact, it facilitated red blood cells lysis with several hemolysins, potentially contributing to blood meal digestion and reduced oocyst load of *Plasmodium* [57], though its effects on *Aedes* were more unclear [58].

On the other hand, *Serratia*, known to have negative or neutral effects on *Aedes* species, was less abundant in NW than in W. For example, in *Ae. aegypti*, this bacterium seems to influence the blood-feeding behavior [59], to reduce the body size of adult males and to increase the development time [54]. In *An. stephensi* Liston, 1901, *Serratia* can even be lethal in certain conditions [59]. However, *Serratia* does not necessarily always have negative or neutral effects on mosquito larvae, as shown by Martinson and Strand [59], that experimentally showed that this bacterium can support *Ae. aegypti* larval development. Hence, the effects of particular

bacterial genera can be also context-dependent. Another bacteria genus, *Hydrogenophaga*, was found to occur only in W, and it is known to have positive effects on certain Chironomidae as a detoxifying agent [60], though its role in mosquito is unknown. Furthermore, we have found *Comamonas* (known to have positive effects on mosquitos, see above) to be abundant in the larval diet, with similar abundance between W and NW individuals. It would be interesting to investigate if the amount of this bacterium in the lab-reared insects is optimal or if it might be advantageous to adopt measures suitable to increase it.

We also found that the bacterial community of W-eggs was the richest in ASVs, while dominance and diversity did not differ among groups. This suggests that lab-reared immatures did not suffer a reduction in the diversity of their microbiota, compared with natural populations. Though we have few data on eggs, our analysis suggests an important difference between W- and NW-eggs. Chen et al. [14] showed that eggs of *Ae. albopictus* contain bacteria which are also main components of gut microbiota of female adults. Hence, having been laid in different environments (wild vs. lab), the microbiota of eggs is different, and may partially affect that of subsequent stages. Because bacterial richness was similar in W- and NW-larvae and pupae, the highest richness found in W-eggs may depend on a component of bacteria in the wild that it is not transferred further across stages.

## Conclusions

Overall, our results indicate that the larval diet which is currently employed for *Ae. albopictus* mass rearing and its management scheme is maintaining a complex bacterial community, including several taxa known to affect the quality of adult mosquitoes positively, though many bacterial effects remain to be tested in our model system. Hence, we think that our study is relevant since our results do not point towards a degradation of bacterial communities in the lab-reared mosquitos, using a promising diet. Because the aim of mass-rearing insects for SIT programs is to produce sterile males of high quality, the effect of adding probiotic substances to the current diet may be explored with specific studies. Certainly one limitation of our study is the sample size, which is not very large. New studies based on a larger sample and on a wider spectrum of rearing regimes and field locations may help to elucidate how the tested diet may affect the microbiota of *Ae. albopictus*.

## Supporting information

**S1 Fig. Rarefaction curves of all the samples.**
(TIF)

**S2 Fig. Histograms showing the SIMPER analysis output performed at genus level of the bacterial community between the groups NW, W and Diet.** Genera are ranked from left to right from the highest to the lowest effect on group discrimination observed in the NMDS. W: Sample collected in the field (wild); NW: Lab-reared sample (non-wild).
(TIF)

**S3 Fig. Stacked barplots showing the relative abundance (abundance %) of the bacterial families (above) and genera (below).** Taxa represented with less than 1% of the total abundance were grouped in the category "other". L: Larvae, P: Pupae, E: Eggs. W: Sample collected in the field (wild); NW: Lab-reared sample (non-wild).
(TIF)

**S1 Table. Samples from eggs, larvae, and pupae of analyzed mosquitoes and yield of DNA extracted.**
(DOCX)

**S2 Table. Descriptive statistics (mean values ± SE) of the 21 genera represented by more than 1% in the bacterial communities analysed, with statistical differences among groups.** W: Sample collected in the field (wild); NW: Lab-reared sample (non-wild). (DOCX)

**S3 Table. Dataset of abundance of taxa across studied samples.** (XLSX)

**S1 Graphical abstract.** (TIF)

## Acknowledgments

We thank Fabrizio Balestrino for the assistance in the initial starting of the study.

## Author Contributions

**Conceptualization:** Carlo Polidori, Paola Mattarelli.

**Data curation:** Carlo Polidori, Luigimaria Borruso, Maria Luisa Dindo, Monica Modesto, Marco Carrieri, Arianna Puggioli, Federico Ronchetti, Romeo Bellini.

**Formal analysis:** Carlo Polidori, Andrea Ferrari, Luigimaria Borruso, Paola Mattarelli, Monica Modesto, Marco Carrieri, Arianna Puggioli, Federico Ronchetti, Romeo Bellini.

**Funding acquisition:** Luigimaria Borruso.

**Investigation:** Carlo Polidori, Andrea Ferrari, Paola Mattarelli, Maria Luisa Dindo, Monica Modesto, Arianna Puggioli, Federico Ronchetti.

**Methodology:** Carlo Polidori, Andrea Ferrari, Luigimaria Borruso, Paola Mattarelli, Maria Luisa Dindo, Monica Modesto, Marco Carrieri.

**Resources:** Romeo Bellini.

**Software:** Andrea Ferrari, Romeo Bellini.

**Supervision:** Carlo Polidori, Paola Mattarelli.

**Writing – original draft:** Carlo Polidori, Paola Mattarelli.

**Writing – review & editing:** Carlo Polidori, Andrea Ferrari, Luigimaria Borruso, Paola Mattarelli, Maria Luisa Dindo, Monica Modesto, Federico Ronchetti.

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
