## [Decision Letter · Decision Letter 0]

5 Jul 2023

PONE-D-23-17566Aedes albopictus microbiota: a comparison between wild and mass-reared immatures fed with a diet based on black soldier fly larvae and fish foodPLOS ONE

Dear Dr. Borruso,

Thank you for submitting your manuscript to PLOS ONE. After careful consideration, we feel that it has merit but does not fully meet PLOS ONE’s publication criteria as it currently stands. Therefore, we invite you to submit a revised version of the manuscript that addresses the points raised during the review process.

We look forward to receiving your revised manuscript.

Kind regards,

Nafiu Bala Sanda, PhD

Academic Editor

PLOS ONE

Journal Requirements:

Additional Editor Comments:

Looking at the comments by three reviewers, I recommend Minor revision.

Reviewers' comments:

Reviewer's Responses to Questions

**Comments to the Author**

1. Is the manuscript technically sound, and do the data support the conclusions?

Reviewer #1: Yes

Reviewer #2: Yes

Reviewer #3: Yes

2. Has the statistical analysis been performed appropriately and rigorously? 

Reviewer #1: Yes

Reviewer #2: Yes

Reviewer #3: I Don't Know

3. Have the authors made all data underlying the findings in their manuscript fully available?

Reviewer #1: Yes

Reviewer #2: Yes

Reviewer #3: Yes

4. Is the manuscript presented in an intelligible fashion and written in standard English?

Reviewer #1: Yes

Reviewer #2: Yes

Reviewer #3: Yes

5. Review Comments to the Author

Reviewer #1: Dear Authors,

The manuscript is written in a very good English and your research was very well structured. The obtained results are very significant from the practical aspect but also represent significant contribution for the science. Sterile Insect Technique depend on the studies as the presented one.

In introduction authors demonstrated relevance and necessity for this study.

Conclusions are in accordance with obtained results.

Considering the Material and method it should be improved by adding details about the samples. It is not very clear what exactly the size of the sample was? How many L1, L2, L3, L4 were used per sample? Or if the authors defined samples differently, please explain or simplify in that section. Also, the number of replicates is not clear.

Please also find below my minor suggestion.

L54 Instead of Non-wild I would prefer to use just lab strain or reared. Because NW does not say much about it.

L57 It should be bacterial classes or classes of bacteria

L316 It is missing who showed because of the reference format (For example [45] showed… there is not author’s name). Same for L207 and L363.

L352 Is “16ositivee” supposed to be positive?

Title: Table S1. Samples from eggs, larvae and pupae of mosquito studied could be modified to

Table S1. Samples from eggs, larvae and pupae of studied mosquitoes… or analyzed mosquitoes.

Why are the numbers in Table S1 approximated and not exact?

Reviewer #2: The research article presented focused on the comparison of the microbiota of non-wild lab-reared Aedes albopictus, wild field-collected Ae. albopictus, and rearing water diet.

The main question is if the microbiota and larval diet have a role in mass rearing programs for SIT.

The authors showed that overall, the microbiota profile significantly differed among the groups of lab-reared Ae. albopictus, wild Ae albopictus, and rearing water diet.

The authors also highlighted the fact that the diet based on Black soldier fly powder and fish food KOI influences the microbiota of non-wild tiger mosquito immature stages, but not in a way that may suggest a negative impact on their quality in SIT programs.

Overall, the manuscript is well written and well structured. The focus of the research is relevant to the field, given the gap of knowledge in the sector.

Figures and tables are clearly presented and correctly labelled, even if some amendments can ameliorate the clarity.

The study design is appropriate to answer the main questions of the study.

Methods are sufficiently detailed to ensure reproducibility of the experiments.

Statistical analyses are relevant and congruous.

However, some improvements are needed in the Results and Discussion sections.

Here are my comments:

- First, I suggest structuring the Results section in paragraphs, describing first the outputs (DNA yields, number of raw sequences, ASVs,…) of the experiments and then presenting the analyses.

- A schematic image could help in understanding the number of samples collected, processed, and then sequenced for each group. It could be presented both in the Results or in the Methods section, depending on the fact that the Authors insert or not the outputs of the experiments.

- As a general observation, the number of samples is low. I understand the difficulty in obtaining samples and a sufficient weight to be processed, but I suggest mentioning this point in the Discussion section.

- In the Results section, please, add DNA yield after DNA extraction, the total number of raw reads obtained and those passed the quality filtering. Specify the number of ASVs obtained.

- I supposed that the total number of sequences obtained was not so high. Is there a relationship with the DNA yield after DNA extraction?

- Do rarefaction curves reach the plateau? Please add this information.

- Fig. 1 legend: please explain the different colours used for dashed lines

- Fig. 2: please adjust x-axis labels.

- Fig. 2 legend: “Note that in few bacterial genera refer to more than one Box-and-whisker plot, because they belong to different SVs.” Please clarify this sentence.

- L316-331: the Authors mentioned different examples of gut bacterial taxa relevant for larval and adult stages, egg production… of different mosquito species. Are those taxa present and relevant also in this study? Add connections to the case study presented if possible. Otherwise, this section seems more suitable for the Introduction.

- L352 correct the typo

- In the Discussion section, in general:

Were the results obtained expected?

Which were the main experimental issues occurred?

What are the next steps to improve the research?

- Conclusion: should be improved by adding a sentence referring to the relevance of the study, before introducing the future perspective of testing probiotics.

- Title: I would modify the title stating the main result obtained instead that what has been done.

Reviewer #3: Manuscript PONE-D-23-17566 by Borruso et al. reports results of a study comparing microbial diversity in immature mosquitoes reared on an artificial diet in the laboratory, in that diet itself, and in wild-collected mosquitoes. This is an important subject that will be interesting to a variety of readers. The study would have benefited from also testing water samples from the habitats where wild mosquitoes were collected, but that would have significantly increased the cost of the study while not being necessary for answering the main research question. Despite these general strengths, I believe that the manuscript needs to be revised. Currently, it is not easy to understand its experimental design. Also, even though it may be my personal idiosyncrasy, using abbreviations for treatments and life stages is unnecessary and makes the reading more difficult.

I also have several more specific comments.

Lines 92-93. Rephrase “mass-rearing a target species to sort the males.”

Line 97. Wild population of which species?

Line 109. Relatively low compared to what?

Line 125. Probably “containing” is a better term than “based.” I presume that probiotics will not be a major component of the diet.

Lines 169-177. Experimental design is not clear from this description. Was it factorial? What were the treatments and what were the replications?

Line 211. Abundances of what?

Lines 216-232. Was that analysis based on the ASVs?

Lines 233-234. Were they deemed to be the most relevant because of their abundance?

Lines 375-377. Perhaps, but this does not really follow from the information presented in the current study, which simply studied the existing diversity.

6. PLOS authors have the option to publish the peer review history of their article (what does this mean?). If published, this will include your full peer review and any attached files.

Reviewer #1: No

Reviewer #2: No

Reviewer #3: No

---

## [Author Response · Author response to Decision Letter 0]

6 Sep 2023

Responses to the REVIEWERS for the ms [PONE-D-23-17566] - [EMID:6bf73daf3d31b61d]

TITLE: Aedes albopictus microbiota: a comparison between wild and mass-reared immatures fed with a diet based on black soldier fly larvae and fish food

Now titled: Aedes albopictus microbiota: differences between wild and mass-reared immatures do not suggest negative impacts from a diet based on black soldier fly larvae and fish food

AUTHORS: Carlo Polidori, Andrea Ferrari, Luigimaria Borruso, Paola Mattarelli, Maria Luisa Dindo, Monica Modesto, Marco Carrieri, Arianna Puggioli, Federico Ronchetti, Romeo Bellini

Reviewer #1

Reviewer #1: General comment: Dear Authors, The manuscript is written in a very good English and your research was very well structured. The obtained results are very significant from the practical aspect but also represent significant contribution for the science. Sterile Insect Technique depend on the studies as the presented one. In introduction authors demonstrated relevance and necessity for this study. Conclusions are in accordance with obtained results.

RESPONSE: We thank the referee for the general largely positive opinion on our ms. All the suggestions of the reviewer were considered and responses can be found below, point-by-point.

Reviewer #1: Considering the Material and method it should be improved by adding details about the samples. It is not very clear what exactly the size of the sample was? How many L1, L2, L3, L4 were used per sample? Or if the authors defined samples differently, please explain or simplify in that section. Also, the number of replicates is not clear.

RESPONSE: Table S1 has been updated to clarify the size and composition of the samples. The table specifies the number of larvae, eggs, or pupae used for each sample. The number of replicates is equal 3 for each sample as described in the table legend and added to the main text. As the amount of sample for the extraction of DNA has been set up at 25 mg (except for field-collected samples where the availability was very low), the table has been modified accordingly.

Reviewer #1: L54 Instead of Non-wild I would prefer to use just lab strain or reared. Because NW does not say much about it.

RESPONSE: we understand the point of the reviewer, but the dichotomy of Wild vs. Non-wild seems quite clear in our opinion. Non-wild insects are easily intended as lab-reared insects, and in the text (Methods) now we further make this more explicit and clear. We thus prefer to keep W and NW. Finally, we stated in a more precise way what W and NW mean in all tables and figures captions.

Reviewer #1: L57 It should be bacterial classes or classes of bacteria

RESPONSE: Thanks for the comment. We have corrected it accordingly.

Reviewer #1: L316 It is missing who showed because of the reference format (For example [45] showed… there is not author’s name). Same for L207 and L363.

RESPONSE: Thanks for the comment; we have corrected it accordingly.

Reviewer #1: L352 Is “16ositivee” supposed to be positive?

RESPONSE: Thanks for the comment; we have corrected it accordingly.

Reviewer #1: Title: Table S1. Samples from eggs, larvae and pupae of mosquito studied could be modified to Table S1. Samples from eggs, larvae and pupae of studied mosquitoes… or analyzed mosquitoes.

RESPONSE: The title of Table S1 has been modified accordingly.

Reviewer #1: Why are the numbers in Table S1 approximated and not exact?

RESPONSE: For low numbers, the exact number has been determined while for number higher than 50 an approximate value has been used. The table S1 has been corrected accordingly.

Reviewer #2

Reviewer #2: General comment: The research article presented focused on the comparison of the microbiota of non-wild lab-reared Aedes albopictus, wild field-collected Ae. albopictus, and rearing water diet. The main question is if the microbiota and larval diet have a role in mass rearing programs for SIT. The authors showed that overall, the microbiota profile significantly differed among the groups of lab-reared Ae. albopictus, wild Ae albopictus, and rearing water diet.

The authors also highlighted the fact that the diet based on Black soldier fly powder and fish food KOI influences the microbiota of non-wild tiger mosquito immature stages, but not in a way that may suggest a negative impact on their quality in SIT programs. Overall, the manuscript is well written and well structured. The focus of the research is relevant to the field, given the gap of knowledge in the sector. Figures and tables are clearly presented and correctly labelled, even if some amendments can ameliorate the clarity. The study design is appropriate to answer the main questions of the study. Methods are sufficiently detailed to ensure reproducibility of the experiments.

Statistical analyses are relevant and congruous. However, some improvements are needed in the Results and Discussion sections.

RESPONSE: We thank the referee for the general largely positive opinion on our ms. All the suggestions of the reviewer were considered and responses can be found below, point-by-point.

Reviewer #2: First, I suggest structuring the Results section in paragraphs, describing first the outputs (DNA yields, number of raw sequences, ASVs,…) of the experiments and then presenting the analyses.

RESPONSE: Thanks for the comment. We have added the required information at the beginning of the Result section.

Reviewer #2: A schematic image could help in understanding the number of samples collected, processed, and then sequenced for each group. It could be presented both in the Results or in the Methods section, depending on the fact that the Authors insert or not the outputs of the experiments.

RESPONSE: Table S1 has been updated to clarify the size and composition of the samples. The table specifies the number of larvae, eggs, or pupae used for each sample. The number of replicates is equal 3 for each sample as described in the table legend and added in the main text. We prefer to add this information in the text rather than adding a new figure (there are already 3 figures and 2 tables).

Reviewer #2: As a general observation, the number of samples is low. I understand the difficulty in obtaining samples and a sufficient weight to be processed, but I suggest mentioning this point in the Discussion section.

RESPONSE: we have added now a sentence at the end of the Conclusion section dealing with this limitation.

Reviewer #2: In the Results section, please, add DNA yield after DNA extraction, the total number of raw reads obtained and those passed the quality filtering. Specify the number of ASVs obtained.

RESPONSE: Thanks for the comment. We have added the required information at the beginning of the result section.

Reviewer #2: I supposed that the total number of sequences obtained was not so high. Is there a relationship with the DNA yield after DNA extraction? 

RESPONSE: As added in the MS after bioinformatics pipelines and quality filtering, a total of 776,170 bacterial reads were found, with an average of 38, 808 ± 13,544 per sample. We did not find any relationship between DNA yield and the number of reads. Actually, this is generally expected; the primary factor influencing the number of reads in Illumina sequencing is indeed the efficiency and quality of the library preparation process. While DNA yield can indirectly impact the library preparation process, it's not the only determining factor.

Reviewer #2: Do rarefaction curves reach the plateau? Please add this information.

RESPONSE: All the samples reached the plateau. A plot showing the rarefaction curves has been added as Supporting information in Fig. S1.

Reviewer #2: Fig. 1 legend: please explain the different colours used for dashed lines

RESPONSE: Done. 

Reviewer #2: Fig. 2: please adjust x-axis labels.

RESPONSE: Done. 

Reviewer #2: Fig. 2 legend: “Note that in few bacterial genera refer to more than one Box-and-whisker plot, because they belong to different SVs.” Please clarify this sentence.

RESPONSE: Done.

Reviewer #2: L316-331: the Authors mentioned different examples of gut bacterial taxa relevant for larval and adult stages, egg production… of different mosquito species. Are those taxa present and relevant also in this study? Add connections to the case study presented if possible. Otherwise, this section seems more suitable for the Introduction.

RESPONSE: We cited a number of studies about the relevance/role/function of certain bacteria that we found in our sample, in order to open to possible hypotheses to test in the future. We think that this part should be kept here, since it refers directly to the results (the bacteria found here). In any case, we explicitly say now that these parts of the discussion serve to test future hypotheses.

Reviewer #2: L352 correct the typo

RESPONSE: corrected

Reviewer #2: In the Discussion section, in general: Were the results obtained expected? Which were the main experimental issues occurred? What are the next steps to improve the research?

RESPONSE: We added several new points in the Discussion related with the questions raised by the reviewer. In particular, limitations of the lack of knowledge of various bacteria’ exact function in our studied species and possible future steps are now highlighted.

Reviewer #2: Conclusion: should be improved by adding a sentence referring to the relevance of the study, before introducing the future perspective of testing probiotics.

RESPONSE: we have now included such a sentence.

Reviewer #2: Title: I would modify the title stating the main result obtained instead that what has been done.

RESPONSE: we have now changed the title according with the reviewer’s suggestion.

Reviewer #3

Reviewer #3: General: Manuscript PONE-D-23-17566 by Borruso et al. reports results of a study comparing microbial diversity in immature mosquitoes reared on an artificial diet in the laboratory, in that diet itself, and in wild-collected mosquitoes. This is an important subject that will be interesting to a variety of readers. The study would have benefited from also testing water samples from the habitats where wild mosquitoes were collected, but that would have significantly increased the cost of the study while not being necessary for answering the main research question. Despite these general strengths, I believe that the manuscript needs to be revised. Currently, it is not easy to understand its experimental design. Also, even though it may be my personal idiosyncrasy, using abbreviations for treatments and life stages is unnecessary and makes the reading more difficult.

RESPONSE: We thank the referee for the general positive opinion on our ms and for the suggestions on how to improve it. All the suggestions of the reviewer were considered and responses can be found below, point-by-point.

 Reviewer #3: Lines 92-93. Rephrase “mass-rearing a target species to sort the males.”

RESPONSE: Thanks for the comment, we have rephrased the sentence.

Reviewer #3: Line 97. Wild population of which species?

RESPONSE: Actually the SIT application is most advanced on Aedes albopictus and Aedes aegypti, but the technology is also under development targeting other mosquito species in the Aedes, Anopheles and Culex genera.

The sentence in the text has been modified accordingly

Reviewer #3: Line 109. Relatively low compared to what?

RESPONSE: we changed “relatively” with “overall”.

Reviewer #3: Line 125. Probably “containing” is a better term than “based.” I presume that probiotics will not be a major component of the diet.

RESPONSE: Thanks for the comment, we have corrected it accordingly.

Reviewer #3: Lines 169-177. Experimental design is not clear from this description. Was it factorial? 

RESPONSE: the experimental design has been clarified.

Reviewer #3: What were the treatments and what were the replications?

RESPONSE: The table S1 has been updated: treatments and replications have been highlighted. Also the main text has been modified accordingly.

Reviewer #3: Line 211. Abundances of what?

RESPONSE: abundance of the bacterial classes, in this sentence. We changed as “of the bacterial classes”.

Reviewer #3: Lines 216-232. Was that analysis based on the ASVs?

RESPONSE: Yes, the analysis was based on ASV and reported in the MM section 

Reviewer #3: Lines 233-234. Were they deemed to be the most relevant because of their abundance?

RESPONSE: we changed “relevant” with “abundant”

Reviewer #3: Lines 375-377. Perhaps, but this does not really follow from the information presented in the current study, which simply studied the existing diversity.

RESPONSE: Here, we just open to such possibility as a further step of research in the future. Adding probiotics is a new step, but knowing prior to experiments how microbiota appears in lab-reared individuals (which is our result) is in our opinion an important starting point.

---

## [Editor Report · Decision Letter 1]

12 Sep 2023

Aedes albopictus microbiota: differences between wild and mass-reared immatures do not suggest negative impacts from a diet based on black soldier fly larvae and fish food

PONE-D-23-17566R1

Dear Dr. Polidori C.,

We’re pleased to inform you that your manuscript has been judged scientifically suitable for publication and will be formally accepted for publication once it meets all outstanding technical requirements.

Kind regards,

Nafiu Bala Sanda, PhD

Academic Editor

PLOS ONE

Additional Editor Comments (optional):

Having gone through the revised manuscript as suggested by the reviewers comments, I have no reservation than to recommend the acceptance of your manuscript for publication in PLOS ONE journal. Congratulations!.
---

## [Editor Report · Acceptance letter]

18 Sep 2023

PONE-D-23-17566R1 

*Aedes albopictus* microbiota: differences between wild and mass-reared immatures do not suggest negative impacts from a diet based on black soldier fly larvae and fish food 

Dear Dr. Borruso:

I'm pleased to inform you that your manuscript has been deemed suitable for publication in PLOS ONE. Congratulations! Your manuscript is now with our production department. 

Kind regards, 

on behalf of

Dr. Nafiu Bala Sanda 

Academic Editor

PLOS ONE